# Sarcopenia Is Associated with Cognitive Impairment Mainly Due to Slow Gait Speed: Results from the Korean Frailty and Aging Cohort Study (KFACS)

**DOI:** 10.3390/ijerph16091491

**Published:** 2019-04-27

**Authors:** Miji Kim, Chang Won Won

**Affiliations:** 1Department of Biomedical Science and Technology, College of Medicine, East-West Medical Research Institute, Kyung Hee University, Seoul 02447, Korea; mijiak@khu.ac.kr; 2Elderly Frailty Research Center, Department of Family Medicine, College of Medicine, Kyung Hee University, Seoul 02447, Korea

**Keywords:** sarcopenia, slow gait speed, cognitive impairment, older adult, aging

## Abstract

Sarcopenia and cognitive impairment may share common risk factors and pathophysiological pathways. We examined the association between impairments in specific cognitive domains and sarcopenia (and its defining components) in community-dwelling older adults. We analyzed 1887 patients who underwent cognitive function tests and dual-energy X-ray absorptiometry from the baseline data of adults aged 70–84 years obtained from the Korean Frailty and Aging Cohort Study. Those with disability in activities of daily living, dementia, severe cognitive impairment, Parkinson’s disease, musculoskeletal complaints, neurological disorders, or who were illiterate were excluded. Cognitive function was assessed using the Korean version of the Consortium to Establish a Registry for Alzheimer’s Disease Assessment Packet, the Frontal Assessment Battery. For sarcopenia, we used the diagnostic criteria of the Asian Working Group for Sarcopenia. The prevalence of sarcopenia was 9.6% for men and 7.6% for women. Sarcopenia (odds ratio [OR] 1.76, 95% confidence interval [CI] 1.04–2.99) and slow gait speed (OR 2.58, 95% CI 1.34–4.99) were associated with cognitive impairment in men. Only slow gait speed (OR 1.88, 95% CI 1.05–3.36) was associated with cognitive impairment in women. Sarcopenia is associated with cognitive impairment mainly due to slow gait speed. Our results suggested that cognitive impairment domains, such as processing speed and executive function, are associated with sarcopenia-related slow gait speed.

## 1. Introduction

The rate of cognitive impairment among older adults without dementia has been estimated at 60 cases per 1000 person-years [1]. Cognitive impairment without dementia is related to a higher risk of progression to dementia and contributes to greater disability, higher neuropsychiatric symptoms, and higher healthcare costs [2,3,4,5]. Cognitive impairment has been reported to contribute to the risk of functional decline in non-disabled people aged 70 years and older [6]. Therefore, identifying risk factors associated with cognitive impairment is important to reduce risks that are potentially modifiable and amenable to interventions. Several risk factors for age-related cognitive decline and impairment have been identified, including lower level of education, cardiovascular risk factors, lifestyle factors, depression symptoms, sleep disorders, traumatic brain injury, inflammatory markers, and related outcomes [7,8].

Sarcopenia is related to loss of skeletal muscle mass and muscle function [9], which represents an important public health issue due to its close link to frailty [10]. Sarcopenia is now formally recognized as a muscle disease with an ICD-10-CM diagnosis code [11]. Age-related sarcopenia has been defined by the European Working Group on Sarcopenia in Older People (EWGSOP and EWGSOP2) based on the combination of appendicular skeletal muscle mass (ASM), muscle strength, and physical performance [9,12]. The EWGSOP2 updated the original definition of sarcopenia to reflect the scientific and clinical evidence that have accumulated over the last decade [12]. Similarly, the Asia Working Group for Sarcopenia (AWGS) [13] and the Foundation for the National Institutes of Health (FNIH) Sarcopenia Project [14] proposed defining sarcopenia as loss of both ASM and muscle function. The estimated prevalence of sarcopenia varies depending on the diagnostic criteria used, ranging from 9.9% to 40.4% in individuals aged 60 years and older [15]. The coexistence of cognitive impairment and sarcopenia may indicate shared common risk factors and pathophysiological pathways [16]. In epidemiological studies, sarcopenia has been linked to cognitive dysfunction [17,18,19]. A systematic review and meta-analysis of seven studies showed that sarcopenia was associated with cognitive impairment [20]. However, this meta-analysis did not probe any specific subdomains of cognitive function. These results also indicated that the associations differ among populations according to the tools used for assessing cognitive function and methods for measuring muscle mass. Therefore, the present study was performed to examine the associations between impairment in specific cognitive domains and sarcopenia (and its defining components) in community-dwelling older adults enrolled in the nationwide Korean Frailty and Aging Cohort Study (KFACS).

## 2. Materials and Methods

### 2.1. Study Population

The KFACS is a multicenter longitudinal cohort study, with a baseline survey conducted in 2016–2017. Participants were recruited among sex- and age-stratified community residents aged 70–84 years in 10 centers in urban and rural regions throughout South Korea [21]. A total of 3014 participants completed the baseline survey, and after 611 participants were excluded from the initial sample based on body composition assessment and bioelectrical impedance analysis, 2403 participants underwent dual-energy X-ray absorptiometry (DEXA). The final analysis included 1887 participants, after the exclusion of 272 participants who had orthopedic surgical implants (e.g., artificial joints, metal sutures, or metal objects in appendicular body regions identified on DEXA images), 209 participants who were classified as dependent for any of the seven activities on the Korean activities of daily living (ADL) scale, and 33 participants who had missing data for physical and cognitive function assessments, dementia, severe cognitive impairment, Parkinson’s disease, who were illiterate, or other neurological disorders (Figure 1). Clinical Research Ethics Committee of Kyung Hee University Medical Center approved the KFACS protocol (Institutional Review Board [IRB] number: 2015-12-103). This study was exempt from needing approval by the Institutional Review Boards of the Clinical Research Ethics Committee of the Kyung Hee University Medical Center (IRB number: 2018-12-040).

### 2.2. Cognitive Function Assessment

Cognitive function was assessed using the Korean version of the Consortium to Establish a Registry for Alzheimer’s Disease Assessment Packet (CERAD-K) [22,23] and the Korean version of the Frontal Assessment Battery [24]. The CERAD battery was developed to distinguish between normal patients and those with mild or moderate dementia [25]. Cognitive function domains such as word list memory, word list recall, word list recognition, trail making test A, digit span forward, digit span backward, Frontal Assessment Battery, and Mini-Mental State Examination were examined. The global cognitive dysfunction was defined as 1.5 standard deviations (SDs) below the age-, sex-, and education-specific norms of the Korean version of the Mini-Mental State Examination (MMSE-KC).

Cognitive impairment was defined as 1.5 SDs below the age-, sex- and education-matched Korean norms on the following cognitive function tests [22,23,24]: processing speed (trail making test A), executive function (Frontal Assessment Battery), verbal episodic memory (word list recall test), and working memory (digit span backward) [26,27]. Generally, neuropsychological tests consider values 1.5 SDs below those of an age- and education-matched control group [28]. Furthermore, the working group of the Subjective Cognitive Decline Initiative proposed the use of age-, sex-, and education-adjusted standardized cognitive tests to classify mild cognitive impairment or prodromal Alzheimer’s disease (AD) [29].

### 2.3. Diagnosis of Sarcopenia

The AWGS defines sarcopenia based on low muscle strength, low muscle mass, and/or low physical performance [13]. The maximum grip strength was measured twice for each hand using a digital handgrip dynamometer (T.K.K.5401; Takei Scientific Instruments Co., Ltd., Tokyo, Japan), while standing upright with the shoulder in neutral position, arms at the side, and elbow fully extended. Low muscle strength was defined as grip strength < 26 kg in men and <18 kg in women. The ASM was measured using DEXA (Hologic DXA; Hologic Inc., Bedford, MA, USA: and Lunar; GE Healthcare, Madison, WI, USA). ASM was calculated as the sum of the lean mass from both the arms and legs (kg). Low muscle mass was defined as an ASM index (ASM/height^2^) of <7.0 kg/m^2^ in men and <5.4 kg/m^2^ in women. The usual gait speed over 4 m with acceleration and deceleration phases of 1.5 m each was measured using an automatic timer (Gaitspeedmeter, Dynamicphysiology, Daejeon, Korea) [27]. Slow gait speed was defined as a gait speed of ≤0.8 m/s. The EWGSOP2 defines sarcopenia based on a combination of muscle strength, ASM, and physical performance [9]. Sarcopenia is confirmed by the presence of low muscle strength (grip strength < 27 kg in men and <16 kg in women) and low muscle quality (ASM index of <7.0 kg/m^2^ in men and <6.0 kg/m^2^ in women). Severe sarcopenia is considered with low muscle strength, low muscle quantity, and low physical performance (gait speed ≤ 0.8 m/s). Sarcopenia was defined according to the FNIH cut-off for weakness, i.e., a grip strength of <26 kg for men and <16 kg for women, and low muscle mass with a ratio of ASM to body mass index [BMI] < 0.789 for men and <0.512 for women. Sarcopenia with slow gait speed was defined as the presence of low muscle mass, weakness, and gait speed ≤ 0.8 m/s.

### 2.4. Other Measurements

All surveys conducted involved an on-site interview and health examination. Information on smoking status, alcohol consumption, physical activity, education level, living conditions, and medical history was provided by the participants. Comorbidities were hypertension, myocardial infarction, dyslipidemia, diabetes mellitus, peripheral vascular disease, angina pectoris, cerebrovascular disease, congestive heart failure, osteoarthritis, rheumarthritis, osteoporosis, asthma, or chronic obstructive pulmonary disease, which were self-reported conditions, diagnosed by physicians. The participants were interviewed to determine whether self-perceived health was poor, fair, very good, or excellent. Self-perceived health was dichotomized as good (very good or excellent) and poor (poor or fair). Depressive symptoms were assessed using the Korean Version of Short Form Geriatric Depression Scale (SGDS-K) [30]. Nutritional status was assessed using the Korean version of the short-form Mini Nutritional Assessment (MNA-SF) [31]. Energy expenditure estimates (kcal/week) were calculated for various activities, and metabolic equivalent scores were derived using the International Physical Activity Questionnaire. Low physical activity level was defined as <494.65 kcal for men and <283.50 kcal for women, with these values corresponding to 20% of the total energy consumed in a population-based Korean survey of older adults from among the general population [32]. Physical function test was assessed using the Short Physical Performance Battery [33,34] and the Timed Up & Go (TUG) test [35].

### 2.5. Statistical Analysis

The Mann-Whitney U test and the chi-squared or Fisher exact test were used to compare the characteristics of participants according to sarcopenia status. We used multiple logistic regression analysis to investigate the association of cognitive impairment, using three diagnostic criteria of sarcopenia (low muscle mass, weakness, and slow gait speed), with sex. Different confounder adjustment models were constructed as follows: model 1, unadjusted; model 2, adjusted by age and education; and model 3, further adjusted by current smoker status, alcohol intake (≥2–3 times/week), possible malnutrition (MNA score ≤ 11 points), low physical activity (<494.65 kcal/week for men and <283.50 kcal/week for women), BMI (≤18.5, 18.5–24.9, ≥25.0), number of comorbidities, depressive symptoms (GDS score point cut-off 6/7), and self-reported health status (good vs. poor). Multiple linear regression analyses to predict the associations between body component indexes (ASM/height^2^, ASM/BMI, and body fat percentage), grip strength, and gait speed with cognitive function tests were performed (by sex; Appendix A). All analyses were performed using SPSS software (ver. 23.0; SPSS Inc., Chicago, IL, USA), with a two-sided *p*-value of <0.05 considered significant.

## 3. Results

Table 1 shows the characteristics according to sarcopenia status. A total of 163 (8.3%) participants were categorized as having sarcopenia according to the AWGS criteria. Participants with sarcopenia were significantly older (78.5 ± 3.8 years vs. 75.5 ± 3.8 years, respectively; *p* < 0.05) and had lower BMI and lower physical activity compared to participants without sarcopenia. The sarcopenic individuals had a higher prevalence of fair/poor self-perceived health (44.2% vs. 22.1%, respectively; *p* < 0.05), were more likely to have depressive symptoms (35.6% vs. 18.1%, respectively; *p* < 0.05), and had poorer nutritional status (17.2% vs. 6.7%, respectively; *p* < 0.05) than the non-sarcopenic individuals. Moreover, the sarcopenic individuals were more likely to have low physical function. Table 2 shows a summary of the neuropsychological test scores according to sarcopenia status as defined by the AWGS. The sarcopenic individuals showed significantly lower scores of MMSE than the non-sarcopenic individuals. Cognitive dysfunction adjusted for age-, sex-, and education-specific norms was not significantly different between the sarcopenia states, for either sex (*p* > 0.05). However, higher cognitive dysfunction rate was seen in men with sarcopenia than those without sarcopenia; however, this relationship did not hold true for women. Among the men, sarcopenic individuals had significantly lower scores on all cognitive tests than the non-sarcopenic individuals (all, *p* < 0.05), with the exception of the backward digit span test. Among the women, word list recognition, trail making test, digit span forward, and Frontal Assessment Battery results were significantly different between the sarcopenic and non-sarcopenic groups (all, *p* < 0.05). Men with sarcopenia showed significantly greater impairments in processing speed (8.5% vs. 2.0%, respectively; *p =* 0.02) and executive function (19.1% vs. 6.2%, respectively; *p* < 0.001) than their non-sarcopenic counterparts. However, a significant difference was noted between female sarcopenic and non-sarcopenic individuals only in processing speed (10.1% vs. 4.4%, respectively; *p* = 0.42).

The associations between sarcopenia, its defining components, and impairment in different cognitive domains in men are shown in Table 3. The results of multivariate analysis adjusted for potential confounding variables (Model 3) showed significant associations between AWGS-defined sarcopenia (odds ratio (OR) = 3.46, 95% confidence interval (CI) = 1.24–9.65), FNIH-defined sarcopenia (OR 3.00, 95% CI 1.08–8.42) and EPGSOP2-defined sarcopenia (OR 2.75, 95% CI 1.01–7.65) and impaired processing speed. Weakness (OR 3.11, 95% CI 1.22–7.90) or slow gait speed (OR 5.54, 95% CI 1.84–16.64) was associated with impaired processing speed. However, there was no significant association between low muscle mass and impaired processing speed (OR 0.89, 95% CI 0.35–2.21). Similarly, sarcopenic individuals (OR 2.98, 95% CI 1.51–5.89) defined according to the AWGS criteria showed significantly higher likelihood of impairment in executive function than non-sarcopenic individuals. In addition, FNIH-defined sarcopenic slow gait speed (OR 5.06, 95% CI 1.07–23.89) and EPGSOP2-defined severe sarcopenia (OR 4.80, 95% CI 1.17–19.70) were associated with impaired executive function. There were no significant associations between sarcopenia and impairments in working memory or verbal episodic memory. Slow gait speed (OR 3.71, 95% CI 1.43–9.62) was only associated with impaired verbal episodic memory. The associations between sarcopenia, its defining components, and impairment in different cognitive domains in women are shown in Table 4. In the multivariate analysis adjusted for potential confounders (Model 3), there were no associations between impairments in subdomains of cognitive function and sarcopenia according to three diagnostic criteria in women. Slow gait speed was associated with impaired processing speed (OR 4.71, 95% CI 2.00–11.09) and impaired executive function (OR 2.97, 95% CI 1.29–6.83). Subgroup analyses showed no associations of impairments in MMSE (score < 24), cognitive dysfunction, and sarcopenia defined according to the AWGS criteria, in either sex (data not shown in table).

Figure 2 shows the association between sarcopenia and cognitive impairment in men and women. AWGS-defined sarcopenia (OR 1.76, 95% CI 1.04–2.99) and slow gait speed (OR 2.58, 95% CI 1.34–4.99) were associated with cognitive impairment in men. However, only slow gait speed (OR 1.88, 95% CI 1.05–3.36) was associated with cognitive impairment in women.

## 4. Discussion

A large community-based sample of non-disabled older adults for this cross-sectional analysis of data collected at the cohort baseline showed that older male adults with any cognitive impairment have a greater risk of AWGS-defined sarcopenia after adjusting for potential confounding variables, but not older female adults. The association in men was mainly due to poor muscle function manifesting as slow gait speed. Furthermore, the results of this study showed that impairment of processing speed or executive functioning was significantly correlated with sarcopenia, as well as weakness and slow gait speed, in men but not in women. The study also found that the presence of slow gait speed was the most sensitive indicator of cognitive impairment, such as executive functioning and processing speed in both sexes.

In this study, the prevalence of sarcopenia in non-disabled older Korean adults was 8.3%, as defined according to the AWGS criteria [13], 13.0% using the EPGSOP2 criteria [9], and 6.0% according to the FNIH criteria [14]. Moreover, prevalence of 1.4% and 1.2% were found when the EWGSOP2 and FNIH definitions of severe sarcopenia were used, respectively. In a systematic review, the pooled prevalence of sarcopenia was shown to range from 9.9% to 40.4% in community-dwelling older adults, depending on the definition used [15]. The estimates were 12.9% (95% CI 9.9–15.9%) using the EWGSOP/AWGS definitions and 18.6% (95% CI 11.8–25.5%) using the FNIH definitions. The low prevalence of sarcopenia and severe sarcopenia in the study sample may indicate that it is an exception to patients with disabilities that affect ADL.

An association between sarcopenia and impaired cognition after adjustment for relevant confounders was also shown by an earlier meta-analysis of six cross-sectional studies [20]. However, this meta-analysis indicated that the associations differed in sub-groups according to the cognitive function tests and body composition measuring devices used in the evaluations. For example, the association of MMSE score with sarcopenia (OR 2.67, 95% CI 1.72–4.16) was stronger than that of the Short Portable Mental Status Questionnaire score (OR 0.97, 95% CI 0.73–1.29). Kim et al. [36] reported that EWGSOP-defined sarcopenia by using bioimpedance spectroscopy was significantly associated with impairment on the MMSE (score < 24) among older adults with end-stage renal disease. In the present study, subgroup analyses showed no associations of impairments on the MMSE adjusted by age-, sex-, and education-specific norms, and sarcopenia according to the AWGS criteria (in either sex), after adjusting for potential confounding variables (data not shown). A previous study that did not support our finding proved an association between MMSE-defined global cognitive dysfunction and sarcopenia based on the AWGS criteria in community living older Taiwanese subjects [37]. Cognitive impairment has been diagnosed in the presence of deficits in one or more neuropsychological tests [28,29,38,39]. Previous studies have found that the MMSE has low sensitivity to detect cognitive impairment and dementia related to probable or possible AD [38,40,41]. Moreover, the MMSE should not be used as a diagnostic replacement for comprehensive neuropsychological assessment [42]. Therefore, this study investigated the potential association between cognitive impairment determined using neuropsychological measures of cognitive domains and sarcopenia, in community-dwelling older adults.

In our study, the association of the presence of deficits on one or more neuropsychological tests, and AWGS-defined sarcopenia was principally impact driven by slow gait speed in men. We also found a slight but not significant association between cognitive impairment and weakness in men (OR 1.54, 95% CI 0.96–2.48, *p* = 0.074). However, cognitive impairment was associated only with slow gait speed in women. Furthermore, in examination of the associations between impairment in specific cognitive domains and sarcopenia (and its defining components), we found that the impairments in processing speed and executive function were significantly associated with sarcopenia, as well as weakness and slow gait speed, in men but not in women. These results were consistent with FNIH-defined sarcopenia and EWGSOP2-defined sarcopenia. Our findings regarding the association between cognitive impairment and slow gait speed are consistent with those of previous studies [43,44,45,46]. The slow gait speed was considered a noninvasive risk factor for cognitive decline in cognitively normal older adults [44]. In longitudinal population-based study, Callisaya et al. [45] presented evidence of the relative contribution of brain atrophy and the progression of white matter lesion volume to gait decline in 225 individuals aged 60–86 years (mean age, 71.4 ± 6.8 years). In this study, the specific cognitive domains showed that the presence of slow gait speed was the most sensitive indicator associated with impairments of processing speed or executive function in both sexes, which was consistent with previous studies [47,48,49,50]. Our sub-analyses showed that all cognitive function test scores significantly positively correlated with gait speed score, with the exception of forward and backward digit span test scores, and word list recognition in women. Doi et al. [51] found that the backward digit span was not significantly correlated with normal walking speed among older adults with MCI. These observations suggested that the smaller volume of the prefrontal area contributes to slow gait speed through slower information processing speed. Furthermore, the association of the prefrontal area with gait speed cannot be explained by cognitive tests of language, memory, and mood [52]. The findings of our study suggest that sarcopenia-related cognitive impairment was primarily mediated by processing speed and executive function among non-disabled older adults. Additionally, gait initiation is a complex transition phase of gait that may induce postural instability [53]. Postural control is the integration of dynamic sensorimotor processes and cognitive processing and is an essential component of postural orientation and equilibrium [54]. Boripuntakul et al. [53] reported that older adults with MCI have reduced postural control when undertaking a challenging walking task such as gait initiation. Processing speed is precise construct as basic cognitive function [55]. Caetano et al. [56] found that processing speed is important for planning and adjusting steps in tasks requiring rapid gait adjustments. It appears particularly important for precise foot placement.

A previous review showed that grip strength decline may be a predictor of loss of cognitive function with aging [57]. Our study did not find an association between AWGS-defined weakness and cognitive impairment. Pentikäinen et al. [58] reported no association between total CERAD score and handgrip strength (β = 0.04, *p* = 0.46) adjusted for age in patients aged 55 to 74 years. Moreover, a population-based longitudinal study of 2288 patients aged 65 years or older [59] found that poor handgrip strength was related to increased risk of dementia in patients with possible MCI. However, poor grip strength was not associated with increased risk of dementia in subjects without apparent cognitive impairment, whereas slow gait speed and poor balance did show this association. Therefore, weakness may be associated with decline in cognitive function in subjects with a greater degree of cognitive impairment. Alternatively, the heterogeneity of results among studies may have been due to differences in the study populations. Our study participants were healthy individuals with low prevalence rates of sarcopenia and cognitive impairment; also, women tended to have lower prevalence rates than men. We speculate that a relatively large number of patients were excluded because of musculoskeletal complaints.

Results among previous studies regarding the association between cognitive impairment and low skeletal muscle mass have been inconsistent. Skeletal muscle mass detected using bioelectrical impedance analysis was linked with cognitive function in patients aged over 60 years in a cross-sectional study [60]. In addition, a negative correlation between low skeletal muscle mass determined by DEXA and cognitive impairment in older women was reported by a large epidemiological study [61]. In this study, low skeletal muscle mass was not independently related to cognitive impairment in either sex. Similarly, in accordance with a previous study in which AWGS-defined low muscle mass, determined by DEXA, was not associated with any cognitive impairment [37]. A systematic review [20] reported that differences among studies in the association of sarcopenia with cognitive impairment were mainly due to differences of body composition devices. Regarding the heterogeneity of the relationship between sarcopenia and cognitive impairment, the association was stronger with use of bioimpedance to measure body composition versus DEXA; sarcopenia defined by DEXA was not significantly associated with cognitive impairment. Our subgroup analyses showed that no cognitive function test scores were associated with ASM index adjusted for height, except word list learning and recall tests in women. Similarly, no cognitive function test scores were associated with ASM index adjusted for BMI (see Appendix A). In contrast, even after adjusting for potential confounding variables body fat percentage was found to be positively associated with cognitive function tests, such as MMSE, word list memory, word list recall, word list recognition, trail making test A, and digit span backward for men, and with MMSE, word list recognition, trail making test A, and Frontal Assessment Battery for women. The positive associations between body fat and cognitive function test in older populations may explain the superior cognition of overweight and obese individuals. By way of explanation, reserves of the hormones estrogen and testosterone in adipose tissue are speculated to be linked, thereby preventing cognitive impairment [62,63]. Smith et al. [64] reported that highest body fat determined by DEXA had significantly higher levels of executive function in patients aged 70 years and older. Additional longitudinal studies are needed to determine the biological mechanism underlying the relationship between body fat mass and cognitive function in older adults.

Our study was limited by its cross-sectional design and did not obtain evidence of a cause–effect relationship between sarcopenia and cognitive impairment. Furthermore, this study examined subjective cognitive impairment. However, neuropsychological testing batteries are commonly used variables in epidemiologic studies. Finally, our results may not be applicable to other settings and populations, because our study only included non-disabled community-dwelling older adults. The major strengths of this study were as follows. The findings of our study are based on a large, nationally representative sample of well-characterized community-dwelling elderly Korean adults. Additionally, a comprehensive test battery was used, and a substantial range of covariates were included in the analysis.

## 5. Conclusions

The present results showed that cognitive impairment is related to sarcopenia primarily through its association with slow gait speed. Therefore, apparent slow gait speed may be useful to identify cognitive impairment at an earlier stage in non-disabled older adults living in the community. Furthermore, our findings suggest that cognitive impairment domains such as processing speed and executive function are associated with sarcopenia, as well as with slow gait speed and weakness, in older men. Additional longitudinal studies are needed to clarify sex-related differences and associations between sarcopenia and its defining components and cognitive impairment.

## Figures and Tables

**Figure 1 ijerph-16-01491-f001:**
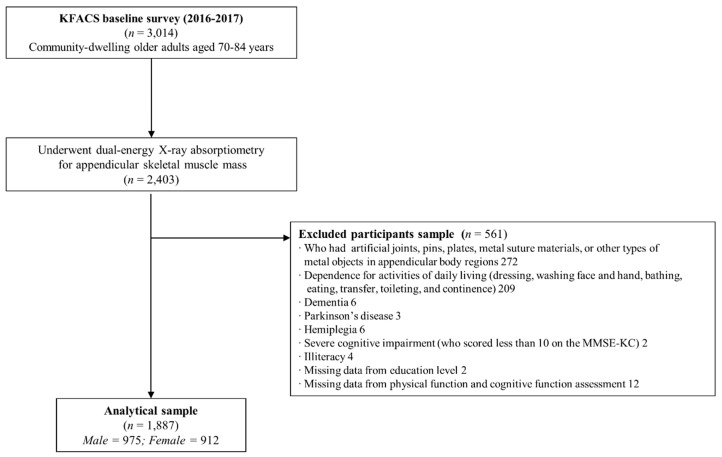
Flow chart of the study population.

**Figure 2 ijerph-16-01491-f002:**
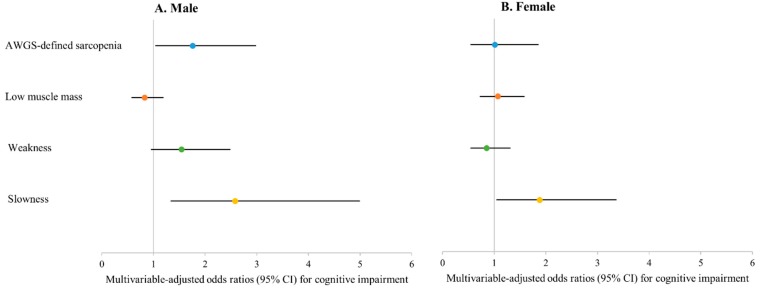
(**A**) Male; (**B**) Female. Associations between sarcopenia, its defining components, and cognitive impairment. Model 3: Adjusted for age, education, smoking status, alcohol intake, Mini Nutritional Assessment Screening score (≤11), low physical activity, body mass index, number of comorbidities, depressive symptoms, and self-reported health status (good vs. poor). Cognitive impairment was defined as a score greater than 1.5 of standard deviation below the corresponding age, sex and education-based Korean norms in any of the four cognitive domains (processing speed, executive function, verbal episodic memory, and working memory).

**Table 1 ijerph-16-01491-t001:** Demographic characteristics of participants with and without sarcopenia by AWGS criteria.

Variable	Overall (*n* = 1887)	Without Sarcopenia (*n* = 1724)	With Sarcopenia (*n* = 163)
Age (years)	75.8 ± 3.9	75.5 ± 3.8	78.5 ± 3.8
Female sex	912 (48.3)	843 (48.9)	69 (42.3)
Education (≥7 years)	1221 (64.7)	1119 (64.9)	102 (62.6)
Live alone	382 (20.2)	343 (19.9)	39 (23.9)
BMI (kg/m^2^)	24.2 ± 2.8	24.4 ± 2.8	22.5 ± 2.9
Appendicular lean mass index (kg/m^2^)	6.46 ± 1.00	6.53 ± 1.00	5.6 ± 0.76
Appendicular lean mass to BMI ratio	0.685 ± 0.157	0.690 ± 0.157	0.632 ± 1.148
Current smoker	119 (6.3)	104 (6.0)	15 (9.2)
Alcohol (≥2–3 time/week)	360 (19.1)	326 (18.9)	34 (20.9)
Low physical activity	174 (9.3)	143 (8.4)	31 (19.3)
Number of comorbidities	1.6 ± 1.2	1.6 ± 1.2	1.9 ± 1.3
Fair/poor self-perceived health	453 (24.0)	381 (22.1)	72 (44.2)
Depressive symptoms (GDS score ≥6)	370 (19.6)	312 (18.1)	58 (35.6)
MNA Screening score (≤11)	144 (7.6)	116 (6.7)	28 (17.2)
**Physical function**			
Handgrip strength (kg)	27.1 ± 7.4	27.7 ± 7.3	21.0 ± 4.9
Usual gait speed (m/s)	1.14 ± 0.25	1.16 ± 0.25	0.96 ± 0.25
Timed get up and go test (s)	10.1 ± 2.4	10.0 ± 2.3	11.8 ± 2.6
Five Times Sit-to-Stand (s)	11.0 ± 3.5	10.8 ± 3.3	13.0 ± 4.6
Short Physical Performance Battery (score)	11.0 ± 1.4	11.1 ± 1.3	10.0 ± 1.8
**AWGS-defined sarcopenia**			
Low muscle mass	709 (37.6)	546 (31.7)	163 (100.0)
Weakness	281 (14.9)	144 (8.4)	137 (84.0)
Slow gait speed	116 (6.1)	68 (3.9)	48 (29.4)

*Notes*: Values are means (± SD) or numbers (percentages). BMI = body mass index; GDS = Geriatric Depression Scale; MNA = Mini Nutritional Assessment. *p*-values are based on the chi-square test, Fisher’s exact test, and the Mann–Whitney U test. Comorbidities were hypertension, myocardial infarction, dyslipidemia, diabetes mellitus, congestive heart failure, angina pectoris, peripheral vascular disease, cerebrovascular disease, osteoarthritis, rheumarthritis, osteoporosis, asthma, or chronic obstructive pulmonary disease, as diagnosed by a physician.

**Table 2 ijerph-16-01491-t002:** Neuropsychological test scores according to sarcopenia by AWGS criteria.

Variable	Male			Female		
	Without Sarcopenia (*n* = 881)	With Sarcopenia (*n* = 94)	*p*-Value	Without Sarcopenia (*n* = 843)	With Sarcopenia (*n* = 69)	*p*-Value
**Global cognitive function**						
Mini-Mental State Examination score	26.5 ± 2.5	25.7 ± 3.1	0.020	25.6 ± 3.2	24.6 ± 3.4	0.021
Mini-Mental State Examination score < 24	148 (16.8)	27 (28.7)	0.007	247 (29.3)	26 (37.7)	0.171
Cognitive dysfunction ^a^	34 (3.9)	6 (6.4)	0.267	27 (3.2)	2 (2.9)	0.890
**Neuropsychological tests**						
Word list learning score	16.9 ± 3.9	14.7 ± 4.3	0.000	17.7 ± 4.2	16.7 ± 4.0	0.078
Word list recall score	5.7 ± 1.9	4.5 ± 2.4	0.000	5.7 ± 2.1	5.5 ± 2.0	0.298
Word list recognition score	8.7 ± 1.7	8.0 ± 2.4	0.006	8.7 ± 1.8	8.3 ± 1.8	0.028
Trail making test A, s	59.8 ± 32.9	82.2 ± 53.9	0.000	86.8 ± 64.9	105.2 ± 79.7	0.025
Digit span forward score	6.3 ± 1.4	6.0 ± 1.5	0.043	5.8 ± 1.4	5.2 ± 1.4	0.001
Digit span backward score	3.6 ± 1.0	3.5 ± 1.2	0.242	3.3 ± 1.1	3.1 ± 1.2	0.234
Frontal Assessment Battery score	14.5 ± 2.5	13.2 ± 3.3	0.000	13.4 ± 2.9	12.7 ± 2.8	0.019
**Cognitive impairment by domains**						
Processing speed	18 (2.0)	8 (8.5)	0.002	37 (4.4)	7 (10.1)	0.042
Executive function	55 (6.2)	18 (19.1)	0.000	65 (7.7)	5 (7.2)	0.561
Verbal episodic memory	77 (8.7)	11 (11.7)	0.217	103 (12.2)	12 (17.4)	0.146
Working memory	44 (5.0)	9 (9.6)	0.060	35 (4.2)	NA	
Cognitive impairment ^b^	161 (18.3)	28 (29.8)	0.013	188 (22.3)	18 (26.1)	0.457

*Notes*: Values are means (± SD) or numbers (percentages). NA = not applicable; *p*-values was based on the Mann-Whitney U test and the chi-square.; ^a^ Cognitive dysfunction was defined as a score greater than 1.5 of standard deviation below the corresponding age, sex, and education-based Korean norms on the Mini-Mental State Examination. ^b^ Cognitive impairment was defined as a score greater than 1.5 of standard deviation below the corresponding age, sex and education-based Korean norms, in any of the four cognitive domains (processing speed, executive function, verbal episodic memory, and working memory).

**Table 3 ijerph-16-01491-t003:** Associations between sarcopenia and impairment of different cognitive domains in men (*n* = 975).

Variable	Model 1	Model 2	Model 3
	Odds Ratio (95% Confidence Interval)		
***Impaired processing speed***			
AWGS-defined sarcopenia			
Low muscle mass (*n* = 485)	1.13 (0.52–2.47)	1.09 (0.98–1.20)	0.89 (0.35–2.21)
Weakness (*n* = 118)	4.08 (1.78–9.34)	3.44 (1.42–8.34)	3.11 (1.22–7.90)
Slow gait speed (*n* = 47)	6.64(2.53–17.43)	5.44 (1.96–15.1)	5.54 (1.84–16.64)
Sarcopenia (*n* = 94)	4.46 (1.88–10.56)	3.74 (1.45–9.61)	3.46 (1.24–9.65)
FNIH-defined sarcopenia			
Sarcopenia (*n* = 76)	3.78 (1.47–9.69)	3.02 (1.10–8.26)	3.00 (1.08–8.42)
Sarcopenia (slow gait speed) (*n* = 11)	8.70 (1.78–42.46)	5.77 (1.09–30.5)	4.47 (0.79–25.69)
EWGSOP2-defined sarcopenia			
Sarcopenia (*n* = 111)	3.65 (1.55–8.61)	3.00 (1.17–7.66)	2.75 (1.01–7.65)
Severe Sarcopenia (*n* = 13)	7.11 (1.49–33.8)	4.87 (0.95–24.92)	3.56 (0.59–21.37)
***Impaired executive function***			
AWGS-defined sarcopenia			
Low muscle mass	1.68 (1.04–2.73)	1.01 (0.95–1.07)	1.11 (0.65–1.90)
Weakness	2.63 (1.48–4.65)	2.64 (1.45–4.83)	2.25 (1.19–4.25)
Slow gait speed	3.20 (1.48–6.90)	3.11 (1.40–6.91)	3.33 (1.39–7.94)
Sarcopenia	3.58 (1.99–6.36)	3.86 (2.04–7.29)	2.98 (1.51–5.89)
FNIH-defined sarcopenia			
Sarcopenia	1.75 (0.84–3.68)	1.68 (0.78–3.63)	1.63 (0.72–3.69)
Sarcopenia (slow gait speed)	4.79 (1.24–18.46)	4.41 (1.10–17.7)	5.06 (1.07–23.89)
EWGSOP2-defined sarcopenia			
Sarcopenia	3.36 (1.93–5.88)	3.65 (1.98–6.73)	2.78 (1.45–5.31)
Severe Sarcopenia	5.75 (1.73–19.15)	5.46 (1.58–18.87)	4.80 (1.17–19.70)
***Impaired working memory***			
AWGS-defined sarcopenia			
Low muscle mass	0.62 (0.39–0.97)	1.06 (1.00–1.13)	0.54 (0.32–0.90)
Weakness	1.73 (0.72–1.92)	1.14 (0.60–2.15)	1.01 (0.58–2.14)
Slow gait speed	2.17 (0.98–4.81)	1.67 (0.74–3.86)	1.69 (0.71–4.00)
Sarcopenia	1.38 (0.71–2.71)	1.07 (0.53–2.18)	1.06 (0.50–2.23)
FNIH-defined sarcopenia			
Sarcopenia	1.20 (0.56–2.06)	0.93 (0.42–2.07)	0.88 (0.39–1.99)
Sarcopenia (slow gait speed)	3.88 (1.01–14.89)	2.58 (0.64–10.41)	2.19 (0.51–9.43)
EWGSOP2-defined sarcopenia			
Sarcopenia	1.12 (0.58–2.20)	0.85 (0.42–1.72)	0.81 (0.39–1.68)
Severe Sarcopenia	1.85 (0.40–8.49)	1.24 (0.26–5.98)	1.01 (0.19–5.29)
***Impaired verbal episodic memory***			
AWGS-defined sarcopenia			
Low muscle mass	1.18 (0.68–2.06)	1.01 (0.94–1.08)	0.88 (0.46–1.67)
Weakness	1.75 (0.86–3.59)	1.96 (0.92–4.16)	1.85 (0.84–4.06)
Slow gait speed	3.36 (1.43–7.90)	3.86 (1.57–9.45)	3.71 (1.43–9.62)
Sarcopenia	2.01 (0.95–4.27)	2.28 (1.01–5.11)	2.00 (0.84–4.75)
FNIH-defined sarcopenia			
Sarcopenia	1.55 (0.64–3.76)	1.67 (0.67–4.19)	1.70 (0.66–4.39)
Sarcopenia (slow gait speed)	3.98 (0.84–18.89)	4.96 (0.98–25.10)	3.89 (0.69–21.78)
EWGSOP2-defined sarcopenia			
Sarcopenia	1.89 (0.92–3.88)	2.15 (0.99–4.67)	1.92 (0.84–4.38)
Severe Sarcopenia	4.37 (0.94–20.31)	3.71 (0.76–18.02)	2.36 (0.44–12.85)

*Notes*: Model 1: Unadjusted; Model 2: Adjusted for age and education; Model 3: Further adjusted for smoking status, alcohol intake, Mini Nutritional Assessment Screening score (≤11), low physical activity, body mass index, number of comorbidities, depressive symptoms, and self-reported health status (good vs. poor).

**Table 4 ijerph-16-01491-t004:** Associations between sarcopenia and impairment of different cognitive domains in women (*n* = 912).

Variable	Model 1	Model 2	Model 3
	Odds Ratio (95% Confidence Interval		
***Impaired processing speed***			
AWGS-defined sarcopenia			
Low muscle mass (*n* = 251)	0.87 (0.43–1.75)	0.96 (1.05–1.23)	0.97 (0.44–2.16)
Weakness (*n* = 163)	2.01 (1.03–3.93)	1.37 (0.68–2.78)	1.19 (0.56–1.23)
Slow gait speed (*n* = 69)	8.79 (4.48–17.24)	4.46 (2.11–9.44)	4.71 (2.00–11.09)
Sarcopenia (*n* = 69)	2.46 (1.05–5.74)	1.82 (0.74–4.48)	1.67 (0.62–4.47)
FNIH-defined sarcopenia			
Sarcopenia (*n* = 38)	0.55 (0.70–3.90)	0.28 (0.04–2.16)	0.24 (0.03–1.99)
Sarcopenia (slow gait speed) (*n* = 11)	2.00 (0.25–15.95)	0.67 (0.80–5.53)	0.55 (0.06–5.23)
EWGSOP2-defined sarcopenia			
Sarcopenia (*n* = 57)	2.53 (1.02–6.26)	1.62 (0.63–4.12)	1.18 (0.41–3.37)
Severe Sarcopenia (*n* = 14)	8.58 (2.58–28.55)	2.91 (0.80–10.57)	1.79 (0.42–7.61)
***Impaired executive function***			
AWGS-defined sarcopenia			
Low muscle mass	0.98 (0.57–1.69)	0.99 (0.93–1.06)	1.05 (0.58–1.91)
Weakness	0.95 (0.50–1.81)	0.95 (0.49–1.84)	0.83 (0.42–1.65)
Slow gait speed	2.85 (1.45–5.61)	3.89 (1.61–7.15)	2.97 (1.29–6.83)
Sarcopenia	0.94 (0.36–2.41)	0.96 (0.37–2.49)	0.97 (0.36–2.61)
FNIH-defined sarcopenia			
Sarcopenia	0.66 (0.16–2.79)	0.66 (0.15–2.85)	0.57 (0.13–2.58)
Sarcopenia (slow gait speed)	2.72 (0.58–12.85)	2.94 (0.59–14.70)	2.27 (0.42–12.37)
EWGSOP2-defined sarcopenia			
Sarcopenia	0.90 (0.32–2.57)	0.91 (0.32–2.63)	0.70 (0.23–2.12)
Severe Sarcopenia	3.38 (0.92–12.42)	3.93 (0.98–15.76)	2.77 (0.61–12.52)
***Impaired working memory***			
AWGS-defined sarcopenia			
Low muscle mass	1.02 (0.66–1.57)	1.04 (0.99–1.09)	1.13 (0.67–1.83)
Weakness	1.42 (0.78–2.61)	1.00 (0.60–1.66)	0.88 (0.52–1.49)
Slow gait speed	2.06 (1.12–3.79)	1.40 (0.73–2.70)	1.49 (0.72–2.90)
Sarcopenia	1.51 (0.79–2.91)	1.37 (0.70–2.69)	1.36 (0.67–2.75)
FNIH-defined sarcopenia			
Sarcopenia	1.60 (0.69–3.73)	1.25 (0.53–2.96)	1.19 (0.48–2.91)
Sarcopenia (slow gait speed)	5.99 (1.78–20.00)	3.85 (1.11–13.37)	3.36 (0.95–12.60)
EWGSOP2-defined sarcopenia			
Sarcopenia	1.52 (0.75–3.10)	1.25 (0.60–2.61)	1.02 (0.47–2.21)
Severe Sarcopenia	5.43 (1.85–15.94)	3.49 (1.12–10.86)	2.70 (0.81–8.96)
***Impaired verbal episodic memory***			
AWGS-defined sarcopenia			
Low muscle mass	0.33 (0.12–0.94)	1.10 (1.01–1.20)	0.33 (0.10–1.07)
Weakness	2.51 (1.22–5.15)	1.87 (0.88–3.69)	1.75 (0.81–3.80)
Slow gait speed	2.12 (0.79–5.64)	0.93 (0.33–2.65)	0.95 (0.31–2.95)
Sarcopenia	NA	NA	NA
FNIH-defined sarcopenia			
Low muscle mass	0.98 (0.64–2.07)	0.88 (0.41–1.90)	0.93 (0.41–2.10)
Sarcopenia	1.42 (0.32–6.13)	0.88 (0.20–3.92)	0.79 (0.16–3.83)
Sarcopenia (slow gait speed)	2.55 (0.32–20.49)	1.01 (0.12–8.49)	0.86 (0.09–8.32)
EWGSOP2-defined sarcopenia			
Sarcopenia	2.15 (0.99–4.67)	1.85 (0.66–5.17)	1.75 (0.57–5.53)
Severe Sarcopenia	4.37 (0.94–20.31)	1.68 (0.33–8.49)	1.43 (0.24–8.60)

*Notes*: Model 1: Unadjusted; Model 2: Adjusted for age and education; Model 3: Further adjusted for smoking status, alcohol intake, Mini Nutritional Assessment Screening score (≤11), low physical activity, body mass index, number of comorbidities, depressive symptoms, and self-reported health status (good vs. poor). NA = not applicable.

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
