# Peer review of "Sarcopenia Is Associated with Cognitive Impairment Mainly Due to Slow Gait Speed: Results from the Korean Frailty and Aging Cohort Study (KFACS)"

_ijerph, 2019, doi:10.3390/ijerph16091491_

Round 1

Reviewer 1 Report

The authors examined the association between Sarcopenia and decline in specific cognitive domains using the CERAD. They found some association between  Sarcopenia and cognitive functioning for individuals with slow gait.

The article needs to have more of a theoretical basis about the cause of the relationship between Sarcopenia and cognitive impairment.  Of course, the authors would have to acknowlege that  one weakness of their design is that it is not longitudinal so that it is difficult to determine what comes first--Sarcopenia or cognitive impairment.  That is, cognitive impairment might result in individuals becoming less active in the environment and might result in these individuals becoming socially isolated.  Social isolation and a sedentary lifestyle have a deleterious effect on cognitive functioning.

In the discussion section, the authors should elaborate on the relation between speed of processing and postural equilibrium.  There is a body of literature that relates speed of processing and executive functioning to gait speed.

Author Response

To the Reviewer

We appreciate this opportunity to address the reviewer’ comments and to revise our manuscript. The authors would like to thank the reviewer for their critical review, comments, and suggestions for improving the quality of our manuscript. The following responses have been prepared to address each comment in a point-by-point fashion. We have indicated revisions in the manuscript with yellow highlights (red text). All page and paragraph numbers refer to locations in the revised manuscript.

Reviewer 2 Report

This is a study based on a well-characterized cohort, aiming to assess associations between cognitive impairment and sarcopenia. The authors provide a good motivation for the study and describe most of the assessments and variables in a sound and detailed manner, butthere a several areas which need improvement, notably in the methods section and the presentation of results.

Major points:

The use of p-values and statistical significance needs to be reviewed, keeping critical discussions of stat. significance in mind (see e.g Am Stat Assoc. statements). Simply judging results by significance does not do justice to some of the associations detected.

Authors should try and provide more guidance to the readers, eg in terms of organizing the discussions. They have arranged the discussion such that all results per measure of sarcopenia are discussed consecutively, and this is somewhat tiring and at times confusing to read. So a short introduction to the way the discussion is organized will be helpful, and also critically go through the text to sharpen the arguments and aim for more clarity and brevity.

Strengths and limitations are not well developed.

Tables should be revised such that unnecessary p-values are deleted and the formating and legends are improved and standardized.

Minor points:

I have made comments directly in the document, including both main and minor comments. I hope this helps the authors to improve the manuscript further.

Author Response

To the Reviewer

We appreciate this opportunity to address the reviewer comments and to revise our manuscript. The authors would like to thank the reviewer for their critical review, comments, and suggestions for improving the quality of our manuscript. The following responses have been prepared to address each comment in a point-by-point fashion. We have indicated revisions in the manuscript with yellow highlights (red text). All page and paragraph numbers refer to locations in the revised manuscript.

Reviewer 3 Report

This is an interesting paper on the relationship between sarcopenia and cognitive status.  The paper uses appropriate statistical tests and controls, is well powered, and asks a question that has utility to the medical community.  

I recommend this paper be published in its present form.

*Minor comment:

Page 1, lines 39-40, where you list "education" as a risk factor, perhaps consider changing the wording to "low education".

Author Response

(The authors gave the same response as above.)

Round 2

Reviewer 2 Report

My comments have been addressed. I accept that re-organizing the discussion may not lead to a better outcome, although a short description of what is covered in the discussion could have been useful - but it is not required.